# Declarative Description of Knowledge Graphs Construction Automation: Status & Challenges

David **Chaves-Fraga**[1,2], Anastasia **Dimou**[1]

[1]*KU Leuven, Department of Computer Science, Sint-Katelijne-Waver, Belgium*

[2]*Universidad Politécnica de Madrid, Campus de Montegancedo, Boadilla del Monte, Spain*

### Abstract

Nowadays, Knowledge Graphs (KG) are among the most powerful mechanisms to represent knowledge and integrate data from multiple domains. However, most of the available data sources are still described in heterogeneous data structures, schemes, and formats. The conversion of these sources into the desirable KG requires manual and time-consuming tasks, such as programming translation scripts, defining declarative mapping rules, etc. In this vision paper, we analyze the trends regarding the automation of KG construction but also the use of mapping languages for the same process, and align the two by analyzing their tasks and a few exemplary tools. Our aim is not to have a complete study but to investigate if there is potential in this direction and, if so, to discuss what challenges we need to address to guarantee the maintainability, explainability, and reproducibility of the KG construction.

### Keywords

Knowledge Graphs, Automation, Explainable AI, Declarative Rules

## 1. Introduction

A lot of works on knowledge graph (KG) construction are focused on defining mapping languages to declaratively describe the transformation process, and on optimizing the execution of such declarative rules. The mapping languages rely on either dedicated syntaxes, such as the family of languages around the W3C recommended R2RML[1] (e.g., RML [1] or R2RML-F [2]), or on re-purposing existing specifications, such as query languages like the W3C recommended SPARQL[2] (e.g., SPARQL-Generate [3] or SPARQL-Anything [4]), or constraints languages like ShEx[3] (e.g., ShExML [5, 6]).

Despite the plethora of mapping languages and the increasing number of optimizations for the execution of the declarative rules, these rules are still defined through a manual and time-consuming process, affecting negatively their adoption. Different solutions were proposed to automate the definition of mapping rules that describe how a KG should be constructed. On the one hand, MIRROR [7], D2RQ [8] and Ontop [9] follow a similar approach, extracting from the RDB schema a target ontology and the mapping correspondences. On the other hand,

*KGCW'22: International Workshop on Knolwedge Graph Construction, May 30, 2021, Creete, GRE*

✉ david.chaves@upm.es (D. Chaves-Fraga); anastasia.dimou@kuleuven.be (A. Dimou)

🆔 0000-0003-3236-2789 (D. Chaves-Fraga); 0000-0003-2138-7972 (A. Dimou)

[1]http://www.w3.org/TR/r2rml/

[2]https://www.w3.org/TR/sparql11-overview/

[3]https://shex.io/

AutoMap4OBDA [10] and BootOX [11] consider an input ontology and generate actual R2RML mappings from the RDB. However, these solutions are focused on declarative solutions only for relational databases, while recent solutions investigate non-declarative automation of KG construction.

Beyond relational databases, the recent SemTab challenge[4] present a set of tabular datasets [12] with the aim of matching them automatically to external KGs, such as DBpedia and Wikidata. The proposed solutions [13, 14, 15] address the problem using different techniques, such as heuristic rules, fuzzy searching over the KGs, or knowledge graph embeddings. Although their final objective is the same (obtain high precision and recall results) and they perform similar procedures, each solution implements its own workflow and addresses each proposed task by SemTab in different ways. Hence, making a fair and fine-grained comparison among the different solutions to understand how they obtain the actual results is not an easy task.

In this vision paper, we align the tasks followed by solutions for the automation of the semantic table annotation with concepts of existing declarative solutions. We indicatively select and analyze a few tools for the automation of KG construction and identify common steps. We discuss if they can be declaratively described relying on existing mapping languages, and what the challenges are to proceed in this direction. We consider the RDF Mapping Language (RML) [1] as a high-level and general representation to describe the schema transformations and its extension, the Function Ontology (FnO) [16] to describe the data transformations.

Our objective is not to present a complete study, but to investigate if there is potential in this direction. By describing the steps followed by different solutions in a more fine-grained and standard manner, we make the steps comparable, and we can better discuss what challenges we need to address to guarantee the maintainability, explainability and reproducibility of the KG construction, as well as to ensure the provenance of each performed task.

## 2. Task alignment with mapping languages

We analyze the different steps of the SemTab challenge, inspect the relation between the SemTab challenge tasks and align them with concepts from the declarative construction of RDF graphs (Figure 1). To achieve this, we include the relationship between each of the tasks and their potential declarations within a mapping language. We considered the RML mapping language because it is commonly used and the authors are more familiar with, but we are confident that the other mapping languages could express the same concepts. Before we proceed with the alignment, we give a small introduction on the SemTab challenge and RML:

**SemTab challenge**   The SemTab challenge consists of three tasks: (i) cell to KG entity matching (CEA), which matches cells to individuals; (ii) column to KG class matching (CTA), which matches cells to classes; and (iii) column pair to KG property matching (CPA), which captures the relationships between pairs of columns.

**RML**   The RDF mapping language (RML), a superset of the W3C recommended R2RML, expresses schema transformations from heterogeneous data to RDF. An RML mapping contains

---

[4]https://www.cs.ox.ac.uk/isg/challenges/sem-tab/

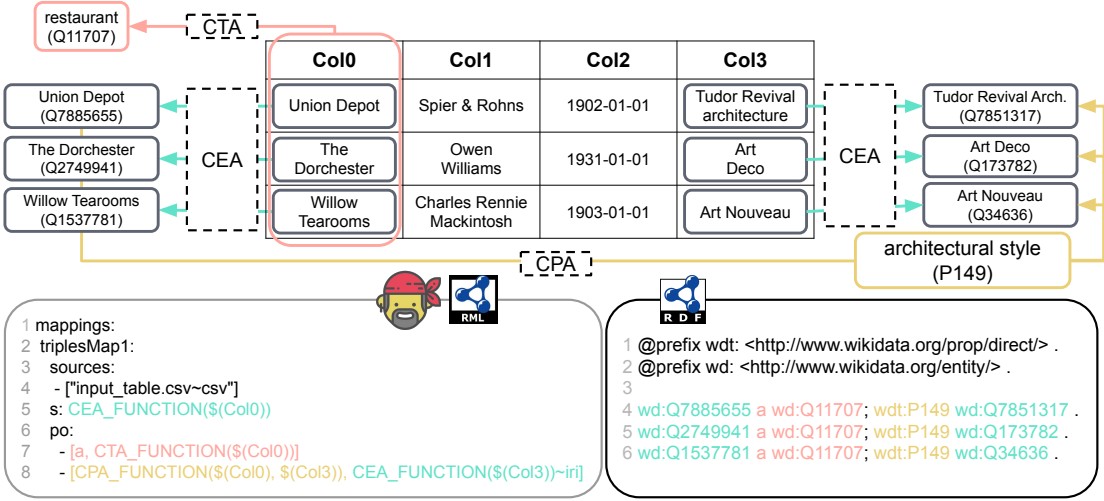

**Figure 1: Automation tasks alignment within declarative mapping language.** Example extracted from SemTab 2021 challenge, where the CEA, CTA and CPA tasks are aligned with a declarative construction of a knowledge graph using the RML mapping language (YARRRML serialisation).

one or more *Triple Maps* which on their own turn contain a *Subject Map* to generate the subjects of the RDF triples, and zero or more *Predicate Object Maps* with pairs of *Predicate* and *Object Maps* to generate the predicates and the objects respectively for each incoming data record. RML was aligned with the Function Ontology (FnO) [16] to describe the data transformations which are required to construct the desired RDF graph, ensuring that the functions are independent from any implementation.

We analyze how the different tasks of the challenge contribute in constructing a part of an RDF triple, and we align these tasks with the corresponding concepts of the RML mapping language that construct the same part of an RDF triple.

**Cell-Entity Annotation (CEA):** This task identifies the URI of an entity from a cell. In the target RDF graph, this is the subject or the object of the RDF triple. In Fig. 1, the `Col0` values are used to obtain the subjects of the triples while the `Col3` values generate the objects (both green colored in the RDF extract of Fig. 1). If a declarative approach is considered to generate these triples, for example in RML, the `rr:subjectMap` property is used (line 5 of RML doc in Fig. 1), which declares how the subjects of the triples are generated and the `rr:objectMap` (line 8 of RML doc in Fig. 1), when the expected objects are in the form of URIs.

**Column-Type Annotation (CTA):** This task predicts the common class of a set of items given a column from the table. SemTab assumes that a table only generates one kind of entity (i.e. the first column is used for CTA). In Figure 1, we can observe that the URIs retrieved using `Col0` are considered for obtaining the corresponding shared concept (i.e., restaurant) (red colored in the RDF extract of Fig. 1). Declaring the class in RML can be done through the

shortcut `rr:class` property within the `rr:SubjectMap` (line 7 of RML doc in Fig. 1) or using a `rr:predicateObjectMap` with a `rdf:type` fixed predicate.

**Columns-Property Annotation (CPA):**   This task aims to predict the property that relates the CTA column (subjects) to the rest of the columns. Fig. 1 shows a CPA task that relates the `Col0` with the `Col3` through the property `architectural style` (`wdt:P149`, yellow colored in the RDF extract). In RML, the predicates of the triples are declared using the `rr:predicateMap` property (line 8 of RML doc in Fig. 1), and unlike typical mapping rules, where it is usually assumed that predicates are constants (as they are declared in the input ontology), the predicates depend on the data, hence they are dynamically defined.

Based on the aforementioned analysis, we conclude that the tasks performed to automate the KG construction can be aligned with concepts from declarative mapping languages. The CEA task is aligned with the *RDF term* construction for the subject or the object of the RDF triple, the CTA task assigns the class and the CPA task aligns with the *Predicate and Object Map*.

## 3. Comparing semantic tabular matching systems

In this section, we analyze in detail the steps performed by some of the tools proposed for solving the SemTab challenge. The comparative analysis among the three selected engines (summarized in Table 1), is not meant to be exhaustive. We aim to identify if there are common steps and functions that the engines perform to accomplish the challenge's tasks and ultimately if it is possible and desired to declaratively describe them with mapping languages.

### 3.1. Selected Systems

We indicatively selected the systems that: (i) obtained good results in the SemTab 2021 challenge[5]; and (ii) have the source code openly available. Therefore, we included in this comparison JenTab [14], MTab [13] and MantisTable V [17]. The use of different terminologies for describing similar tasks (e.g., majority vote in Mantis V is referred as frequency) and the complexity of the proposed workflows, where the results from one of the task influence the others in a iterative way, create difficulties to compare the approaches and reproduce their results.

**JenTab**[6] participated in SemTab 2020 and 2021, and it was always positioned among the top five solutions for most rounds. It follows a heuristic-based approach proposing the CFS (Create, Filter, Select) approach for all tasks and with different configurations and workflows.

**MTab**[7] participated in all SemTab editions, winning the first prize in 2019 and 2020. Apart from the support of multilingual datasets, MTab implements several approaches for performing the entity search (i.e., CEA): keyword search, fuzzy search, and aggregation search[8].

**MantisTable** V[9] is an extended and improved version of MantisTable [18]. Similarly to JenTab, MantisTable has also participated in SemTab 2020 and 2021 editions. It implements a

[5]https://www.cs.ox.ac.uk/isg/challenges/sem-tab/2021
[6]https://github.com/fusion-jena/JenTab
[7]https://github.com/phucty/mtab_tool
[8]https://mtab.app/mtabes/docs
[9]https://bitbucket.org/disco_unimib/mantistable-v/

set of heuristic rules (similar as JenTab) and complex string similarity functions for the entity recognition task (like MTab). Additionally, it provides a general and efficient tool (LamAPI) to fetch the necessary data for all SemTab tasks, independently of the target KG.

## 3.2. Observations

The systems we inspected follow the *same steps*: they perform a preprocessing step, and setup lookup and datatype prediction services. Then the CEA task is performed followed by the CTA and CPA tasks which depend on the CEA task. Given that the systems follow the same steps, we could map the three main tasks (CEA, CPA, CTA) to the Create-Filter-Select (CFS) procedure proposed by JenTab (see Table 1).

We observe similarities in most tasks among the engines. The subtasks performed in the *preprocessing step*, are very similar in the three engines. The preprocessing tasks include several functions, such as fixing encoding issues, removing HTML tags or special characters, and detecting missing white spaces (see Table 1), and they usually delegate them to third-party libraries (e.g., ftfy[10]). We observe similar tasks are performed when declarative solutions are used for cleaning and preparing the data. These preprocessing tasks are described with FnO in the case of RML and executed either together with the schema transformations or as a preprocessing task too.

The same occurs for the *datatype prediction*, where regular expressions are often used to detect if cell values are entities or literals, and what type of literals (string, date, or numbers). In the case of declarative solutions, this datatype inspection task is performed manually. However, adjusting the datatype is possible relying on functions for data transformations.

Most of them also incorporate a lookup step to retrieve the necessary data from the KGs (e.g., using SPARQL queries), including similarity functions or fuzzy search. The *search engine for the KG lookups* in JenTab and Mantis V is ElasticSearch, although the former implements the Jaro Winkler distance [19] while the later embeds it in a more efficient engine and exploits its query capabilities. Lookups were also incorporated in the case of declarative solutions [20], where lookup services retrieve a URI to identify an entity instead of assigning a new one.

As far as the actual tasks is concerned, each engine performs its *own approach* for the CEA, CTA, and CPA tasks, although we also find some similarities. The most important ones that are implemented in the three engines are: (i) the Levenshtein distance [21] for filtering candidates, and (ii) the majority vote (called frequency in Mantis V) for selecting the final annotations. We believe that the use of declarative approaches, such as the Function Ontology [16] for describing common functions (e.g., Levenshtein), could make the solutions more comparable. It would also be clearer if they perform the same function, and more explainable, as current solutions for the automation of KG construction act like blackboxes: neither their implementations are open sourced nor the declarative descriptions of what they execute are available. Providing at least declarative descriptions of the performed tasks would enhance the transparency of these solutions.

---

[10]https://pypi.org/project/ftfy/

**Table 1**
Tasks comparison among different SemTab solutions

| | | JenTab | MTab | Mantis V |
|---|---|---|---|---|
| **KG Lookup** | | ElasticSearch on top of KG SPARQL Queries | WikiGraph Generation Ad-hoc API | LamAPI(ElasticSearch, Mongo and Python) |
| **Preprocessing** | Fix encoding | Y | Y | N |
| | Special characters | Y | N | Y |
| | Restore missing spaces | Y | N | Y |
| | Remove HTML tags | N | Y | N |
| | Remove non-cell-values | N | Y | N |
| **Datatype** | | REGEX Type-based cleaning | Cell values identification (literal, entity) SpaCy models for potential types Majority vote to define column type | REGEX for datatypes exceeding a threshold Entity columns that do not exceed the threshold |
| **CEA** | CREATE | Different query rewriting techniques | Keword search (BM25) Fuzzy search (Levenshtein distance) | LamAPI lookup with IB similarity |
| | FILTER | Levenshtein distance (among others) | Filter and hashing (Symetric Delete) Context similarities by row | Levenshtein confidence score for entities Literals XXX |
| | SELECT | Levenshtein distance | Highest context similarity | XXXX |
| **CTA** | CREATE | Types from CEA | Types from CEA | Types from CEA |
| | FILTER | Remove the less popular types | - | - |
| | SELECT | Majority vote | Majority vote | Majority vote |
| **CPA** | CREATE | Cell annotations (CEA) and fuzzy match for data properties | Aggregate all properties from CEA by row | Properties from CEA lookups |
| | FILTER | - | - | - |
| | SELECT | Majority vote | Majority vote | Majority vote |

# 4. Challenges for a declarative automation of KG Construction

We identify a set of challenges that need to be addressed to declaratively describe solutions for automatic KG construction. These challenges can be divided into two main categories: technical challenges and conceptual challenges.

On the technical side, there is a major difference between the solutions for the automation of KG construction and the execution of declarative KG construction solutions: The solutions for automatic KG construction rely on *iterative processes* that continuously refine and improves a task, while the different tasks influence each other. To the contrary, the declarative KG construction is a *linear process* that is executed only once. Not all declarative rules are executed linearly, solutions that restructure [6] or parallelize them [22, 23] are increasingly encountered. Thus, if the solutions for automatic KG construction are declaratively described, their iterative execution needs to be described as well. How do we do that with the mapping languages?

Besides the overall execution process, the *iteration patterns* are different. The solutions for automatic KG construction are applied to all directions, both per column and per row, and even combined. To the contrary, the declarative solutions are applied only per row, and the mapping languages are designed under this assumption. Should the mapping languages be extended to support more iteration patterns? If so, would the `rml:iteration` for RML and the relevant constructs in the other mapping languages be sufficient or more adjustments are required?

The solutions for automatic KG construction rely on *interrelated tasks* which may produce *intermediate representations*, and their results impact the rest of tasks. Thus, the declarative KG construction solutions need to deal with dynamic and recursive steps (e.g., intermediate representation of the input data sources and mapping rules, multiple function execution, etc.) that can negatively impact the generation process. Hence, declaratively describing is a challenge. Should the mapping languages be further extended then?

On the conceptual side, there are two main differences with respect to the training and target KG. In most real projects that declarative solutions tackle, the input data and sometimes the target ontology are only provided, but there is neither similar data to train the solutions nor existing KGs that can be used to find entities or to predict the relationships. While relying on ontology matching techniques between existing KGs (e.g., DBPedia, Wikidata) and the target ontology or exploiting NLP approaches between ontology and input sources documentation could be a solution for the latter, would it be realistic given that most ontologies are not aligned and not all of them provide documentation?

## 5. Conclusions and Future Work

In this paper, we analyze the KG construction solutions and compare the automatic with the declarative. While the tasks can be aligned with respect to what they achieve, their execution is fundamentally different and a direct alignment is not feasible.

Automatic solutions for KG construction are required to facilitate the adoption of KGs, but there are also merits when the automation tasks are declaratively described, with respect to maintainability, sustainability, and reproducibility. However, directly aligning the automatic solutions with the declarative solutions might be technically and conceptually challenging considering their different execution and iteration patterns. Extending the existing mapping languages would be a solution, but it would also require to address the identified challenges and not only. Would such extensions be feasible and desired or would they lead them beyond their purpose? Although, mapping languages are not the only approach to have declarative descriptions. Declarative descriptions of workflows emerge as well. Would that be a more viable solution? If so, would the automatic and declarative solutions keep on growing in different directions? These are questions that would be nice to reflect and discuss during the workshop.

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
