# OpenReview forum: "Declarative Description of Knowledge Graphs Construction Automation: Status & Challenges"
_kg-construct.github.io/KGCW/2022/Workshop — KGCW 2022_

### Official Review · ~Heiko_Paulheim1 · 2022-03-25
**Preliminary, but has a potential to spark interesting discussions and future developments**

**Rating:** 7
**Confidence:** 4

**Review:**

The paper discusses the potential of creating knowledge graphs from tabular data with declarative methods. For that, it compares the techniques used in recent submissions to the SemTab challenge and extracts common operations and differences. Since it is submitted as a position paper, it does not contain an evaluation yet.

Overall, I like the idea of looking from both ends - the systems which exist and the declarative rules. What I would like to have seen is a minimal worked RML example which works on the SemTab dataset, showing which accuracy or the like can be achieved with such a minimal approach. This would also help understanding which gaps exist - which additional mechanisms are missing to RML-based systems, i.e., where they fall behind tailored monolithic table interpretation solutions. It would also be interesting to showcase a formulation of one of the systems (or all three) in RML to the extent possible, with a discussion of which aspects of the systems cannot be expressed in RML.

The example in Fig. 1 leaves me a bit puzzled. It looks like RML can use user defined functions (called CEA_FUNCTION etc.) here, so in the end, the whole endevaour seems to boil down to refactoring the code of the existing approaches so that they can be integrated in a RML pipeline, but since the whole functionality is then hidden in those function, invoked by a rather simple skeleton, what would be the ultimate advantage here? I guess one of those advantages could be that mix-and-match experimentation would become possible (i.e., use CEA from system 1 combined with CTA from system 2), but maybe there are others as well?

In sum, the paper asks a lot of interesting questions, and I could imagine that it will spark a lot of interesting discussions at the workshop.

---

### Official Review · ~Francois_Scharffe1 · 2022-03-31
**On the path from knowledge graph matching techniques towards mapping specifications**

**Rating:** 8
**Confidence:** 5

**Review:**

Importance of the problem
Being able to generate RML mapping rules using automated KG construction techniques is an important step towards the adoption of both KG construction mapping languages and automation techniques. Automating the process reduces the amount of manual work, and on the other hand producing mapping rules facilitates the validation and editing of the mappings through using them in a standard-based user interface.

Scope of the paper
The Semtab challenge is a matching task, proposing to retrieve existing triples from an external KG and match these to data in the tables. KG construction is a more larger task where there is not necessarily any external KG to be matched, or for which only part of the data present in the table is available in the KG (this situation creates a KG completion task). Similarly in the general case the ontology (types and properties) may be given or not, although practically it is reasonable to assume it is a given. It would be useful to discuss these difference in the paper as it would precise the context of what exactly is studied in the more general context of KG construction. This is partially discussed at the end of section 4.

Additionaly it may be good to elude from the discussion the automated generation of knowledge graphs from text, an active research area (see for example the AKBC conference series).

"(line 7 of RML doc in Fig. 1)" does not contain what is specified "rr:class property within the rr:SubjectMap" but "[a, CTA_FUNCTION($(Col0))]". For more clarity you should provide an extension of that figure that materialize the results of the C*A functions into RML.
"In the case of declarative solutions, this datatype inspection task is performed manually. However, adjusting the datatype is possible relying on functions for data transformations." This is a bit out of context in this section as it seem unrelated to the analyzed SemTab engines, isn't it?

Regarding the discussion in Section 4. The declarative language could focus on the result of the matching process. The mapping specification would focus on types and relationships (CTA, CPA) detected by the matching process, and leave Entity mappings to a reference to the result of the matching process (ie match(columnA)). This way the inner workings of the -iterative- matching process would be avoided and this would limit the amount of necessary extensions to the mapping language.

---

### Official Review · ~Nora_Abdelmageed1 · 2022-03-31
**Possibilities and challenges regarding the transformation from automatic to declarative KG construction**

**Rating:** 7
**Confidence:** 3

**Review:**

The vision paper discusses possible ways to automate the knowledge graph construction.
The authors establish a comparison baseline between two categories of approaches 1) automatic ways and 2) declarative methods.
In the paper, three of the most recent systems that tackle the semantic table annotation (STA) tasks automatically namely, MantisTable V, MTab, and JenTab have been compared to show the similarities and differences among them. Those systems are three of the core SemTab challenge participants.
In addition, the authors suggested a possible technique to convert such STA into declarative mappings.
Indeed, the paper lacks a complete evaluation but, I think it is ok for a visionary paper.
The authors listed some challenges for such mapping between the automatic and declarative methods, and point out interesting questions that would be worth discussing during the workshop.

---

### Official Review · ~Ernesto_Jimenez-Ruiz1 · 2022-04-06
**Good vision paper to move towards automating KGC**

**Rating:** 8
**Confidence:** 5

**Review:**

The paper presents some good ideas to, one the one hand, provide more automation when defining declarative mappings, and on the other hand to extend the output of state-of-the art Tabular to KG matching systems so that they can also generate declarative mapping to transform the data.

I believe that a language like RML may be fine/enough for now, but it is missing a connection between the annotation of tabular data with a KG  (as in the SemTab challenge) and the transformation of the tabular data into triples using that annotation.

The annotation process is iterative by nature, but I'm not sure if this iterative process should be included within the mappings. I would give the final results of the CEA, CTA and CPA to the mapping definition component. Of course, generated mappings should be reviewed and refined too.

A potential new task for SemTab or another challenge could be evaluating systems not according to the annotations but according to the results of a given set of queries over the generated KG, that is, evaluating the quality of the generated triples out of the tabular data. This is something similar to what we attempted in the RODI benchmark (https://github.com/chrpin/rodi).

To predict the datatype of a column the systems ptype is a good option: https://github.com/alan-turing-institute/ptype

---

### Decision · Program_Chairs · 2022-04-11

**Decision:**

Accept

**Comment:**

Dear authors,

Thank your for submitting your paper. We are happy to inform you that we accept your paper! Please carefully consider the reviews when you prepare your paper for the camera-ready version. You will receive specific instructions to submit your camera-ready soon.

Kind regards
Organizers of the Knowledge Graph Construction workshop 2022